# Design and Development of a Smart Fidget Toy Using Blockchain Technology to Improve Health Data Control

**DOI:** 10.3390/s24206582

**Published:** 2024-10-12

**Authors:** Polina Bobrova, Paolo Perego, Raffaele Boiano

**Affiliations:** Department of Design, Politecnico di Milano, Via Candiani 72, 20158 Milano, Italy; paolo.perego@polimi.it (P.P.); raffaele.boiano@polimi.it (R.B.)

**Keywords:** blockchain, wearable devices, user-centered design, data privacy and control, health data collection

## Abstract

This study explores the integration of blockchain technology in wearable health devices through the design and development of a Smart Fidget Toy. We aimed to investigate design challenges and opportunities of blockchain-based health devices, examine the impact of blockchain integration user experience, and assess its potential to improve data control and user trust. Using an iterative user-centered design approach, we developed a mid-fidelity prototype of a physical fidget device with a blockchain-based web application. Our key contributions include the design of a fidget toy using blockchain for secure health data management, an iterative development process balancing user needs with blockchain integration challenges, and insights into user perceptions of blockchain wearables for health. We conducted user studies, including a survey (*n* = 28), focus group (*n* = 6), interactive wireframe testing (*n* = 7), and prototype testing (*n* = 10). Our study revealed high user interest (70%) in blockchain-based data control and sharing features and improved perceived security of data (90% of users) with blockchain integration. However, we also identified challenges in user understanding of blockchain concepts, necessitating additional support. Our smart contract, deployed on the Polygon zkEVM testnet, efficiently manages data storage and retrieval while maintaining user privacy. This research advances the understanding of blockchain applications in health wearables, offering valuable insights for the future development of this field.

## 1. Introduction

### 1.1. Background

Wearable technologies, like fitness trackers and smartwatches, do more than count steps—they constantly track how much we move, how fast our hearts beat, and even how well we sleep. Wearable devices offer non-intrusive means of gathering data in real time on a range of behavioral and physiological markers associated with mental health [1,2,3].

However, using these wearables also brings several challenges, particularly related to data privacy, data ownership [4], user acceptance, and adoption [2]. To address these challenges, there is growing interest in exploring the potential of blockchain technology [5,6].

Blockchain offers a decentralized and secure way to store and manage data, providing transparency while ensuring privacy and control [7,8,9,10,11]. Blockchain employs advanced cryptographic methods such as hash chaining, anonymous signatures, and non-interactive zero-knowledge proofs to ensure data integrity and privacy [9,12].

Integrating blockchain with IoT systems can address security and privacy challenges by providing decentralized authentication, data integrity, and secure data sharing [13,14]. Blockchain can secure electronic health records by enabling decentralized access control and data confusion, balancing privacy with accessibility [14].

The immutability of blockchain records ensures that once data is recorded, it cannot be altered or deleted, which is crucial for maintaining the integrity of medical records [15,16,17].

Its potential applications in mental health data collection [18,19,20] present an opportunity to overcome the existing barriers and lay the foundation for the efficient use of wearable devices. Distributed ledger technology allows for efficient and secure sharing of electronic health records (EHRs) across different healthcare providers, improving the accuracy and timeliness of diagnoses and treatments [7,15,21].

Blockchain can address interoperability issues by providing a standardized framework for data sharing among different healthcare systems and providers, facilitating seamless access to patient records [7,11,22].

This research explores blockchain technology’s potential to shape wearable devices and UX while enhancing trust. It lies in the overlap of design, technology, and health. The study uses human-centered design to create wearables that focus on developing a smart fidget toy that is conscious of patients’ mental health needs while ensuring that the data gathered is secure, private, and valuable for doctors and therapists.

### 1.2. Motivation

Building upon the exploration of the capabilities of blockchain technology to improve wearable devices for healthcare, our research was narrowed down to focus on a specific device that embodies it: a smart fidget toy. Integrating smart technology into fidget devices offers a unique opportunity to collect mood and health-related data.

A fidget toy is a handheld object designed to help individuals focus, relieve stress, provide sensory experience, or keep their hands busy through repetitive movements [23]. These are used for various applications, notably in aiding individuals with ADHD, autism, and Asperger syndrome [24]. Traditional fidget toys like spinners have been widely adopted for simplicity and playfulness. No peer-reviewed scientific evidence showed that fidgets are effective treatments for mental health conditions [25]. An attempt to address the problem through the use of smart fidgets was proposed by Liang et al. [26] to provide real-time feedback and data tracking for individuals.

However, the potential of integrating smart technology into these toys remains not widely explored, offering opportunities to enhance their functionality and user engagement.

Smart fidget toys have the potential to collect valuable data on user interactions, which can be analyzed to help them build correlations. This data can be used not only to personalize the user experience, making the toys more effective for stress management and focus enhancement, but would also allow to build validated reports for third parties such as doctors, therapists, and researchers and improve collective health information [27].

Data collected from the smart fidget toy can be analyzed by the user and third parties to identify correlations between fidgeting patterns and mood states.

Given the sensitive nature of this data, blockchain technology can be proposed. Blockchain is increasingly being explored for its potential applications in healthcare, offering a decentralized and secure way to manage health-related data. This technology promises to enhance data sharing, maintain patient privacy, and improve the overall efficiency of healthcare systems [28], in particular:Enhanced Data Security: The decentralized nature of blockchain ensures that data is not stored in a single location, reducing the risk of data breaches. Data recorded by the fidget toy can be securely encrypted and stored on the blockchain, making it tamper proof and ensuring that user data remains confidential and unaltered [29].User Control and Privacy: Blockchain empowers users by giving them control over their data. Users can grant or revoke access to their data anytime, ensuring they maintain privacy and have a say in how their information is used. This is particularly important for sensitive health-related data [22,28].Transparency and Trust: The transparency inherent in blockchain technology fosters trust among users. Every transaction is recorded on a public ledger, allowing users to verify the authenticity of their data. This transparency ensures that third parties accessing the data do so with the user’s explicit consent, and any data manipulation attempts can be easily detected [30].Immutable Health Records: Once data is recorded on the blockchain, it cannot be altered or deleted. This immutability is crucial for maintaining accurate health records over time. Users and healthcare providers can rely on the integrity of the data, knowing it reflects accurate historical metrics [31].Efficient Data Sharing: Blockchain facilitates seamless and secure data sharing between users and authorized third parties. For instance, users can share their data with healthcare professionals, researchers, or wellness apps without the need for intermediaries, enhancing the efficiency and speed of data exchange [32].Improved User Engagement: By leveraging blockchain technology, users can be motivated to engage more with the fidget toy and the associated web app. For example, they could earn tokens or rewards for contributing their data to research studies or maintaining consistent usage, adding a layer of gamification and motivation [33]. Moreover, blockchain can improve user engagement by providing security, privacy, transparency, trust, and traceability, as analyzed through user-generated content on Twitter [34].Interoperability: Blockchain can support interoperability with other healthcare systems and applications, ensuring that data stored can be integrated with other health records and analytics platforms. This enhances the value of the collected data and provides a comprehensive view of the user’s health [35].

Despite the promising potential of blockchain-powered fidget toys, their development poses several challenges. To provide users with valuable data, it is necessary to collect quantitative data, such as frequency and duration of fidget toy usage, and qualitative data that captures the user’s emotional state or stress levels by implementing sensor technologies. While integrating blockchain, it is essential to identify the limits of the technology to list the restrictions and requirements for the product, service, and user experience design. Designing a toy that effectively balances traditional tactile satisfaction with digital capabilities requires a thorough understanding of user needs and preferences. The iterative design approach employed in the development of the smart fidget toy was essential in addressing these challenges and ensuring user-centered design.

In this study, we focus on a specific target audience—individuals aged 24 to 32 who are highly interested in well-being practices and are considered technology enthusiasts. This demographic is particularly receptive to new technology, making them an ideal group for testing and refining new smart wearable devices.

Our smart fidget toy is designed for on-demand play, with data collection occurring only when in use. This approach minimizes the computational power required and optimizes memory usage on both the device and the blockchain network. Limiting data collection to active usage periods ensures the device remains efficient and responsive, enhancing the overall user experience.

We assume that the moment the user interacts with our Toy, it is a moment of stress or when they are seeking concentration. Therefore, in the preliminary prototypes, we incorporated only one sensor—a photoplethysmography (PPG) that is effective for sensing the HR and HRV values of healthy subjects during rest [36].

### 1.3. Research Questions and Objectives

This study is part of a broader PhD research project titled User-Centric Design for Health Wearables: Exploring Blockchain Adoption for Data Privacy and Control. The overall aim of the doctoral research is to contribute to the developmental and ethical issues of health wearables and the adoption of blockchain technology. Within this framework, the current study addresses the following primary research questions:What are the design challenges in the design of blockchain-based devices for health?What are the design opportunities in the design of blockchain-based devices for health?

Additionally, we explore secondary questions to provide a comprehensive understanding:How can implementing blockchain technology shape the wearable device, user experience, and related applications?How can blockchain enhance data control, awareness of data value, and user trust?

This study employs a systematic approach that combines user feedback, iterative design processes, technology integration, and validation through user testing and serves as an evaluative single-case study. Through this investigation, we aim to contribute to the interdisciplinary area between design, technology, and health by addressing the design challenges and opportunities of blockchain-based devices for health.

To provide a comprehensive overview of our research framework, Figure 1 illustrates the key components of this study. The primary objective of this research is to explore how iterative design and user-centered techniques can be employed to develop a device collecting health-related data. Through focus group discussions, surveys, and user testing sessions, we seek to understand the key features and functionalities users desire and the most effective ways to implement these features. By documenting our design and development process, we aim to provide valuable insights and guidelines for future design projects incorporating blockchain technology and smart wearables.

### 1.4. Paper Organization

This paper is organized as follows: Section 2 reviews the literature on wearable devices, blockchain in healthcare, and user-centered design. Section 3 describes our methodology, including the iterative design process, user involvement, and development of the prototypes. Section 4 presents results from user studies. Section 5 discusses key findings and challenges in integrating blockchain with wearable health devices. Section 6 concludes the paper, summarizing contributions and future directions. Throughout, we address the research questions and objectives outlined in Section 1.3, focusing on the design and development of our blockchain-based smart fidget toy.

## 2. Literature Review

### 2.1. Wearable Devices in Healthcare

Wearable devices have gained significant interest from experts in healthcare for their ability to monitor physiological and behavioral data continuously. They effectively monitor physiological parameters such as heart rate, blood pressure, and glucose levels, aiding in early diagnosis and treatment [37]. Lu et al. still underline that despite the potential, wearable devices face challenges such as user-friendliness, privacy, and security [37].

Advances in miniaturization, flexible electronics, and biosensors have significantly improved the functionality and reliability of wearable devices [38,39].

Piwek et al. highlight the potential of consumer health wearables to revolutionize healthcare, especially by assisting patients in “self-tracking” [4]. They also underline challenges related to data accuracy and privacy [4].

In the context of mental health, Onyeaka et al. report on the increasing use of smartphones and wearables for health promotion among individuals with anxiety or depression [1]. Schecter et al. discuss the purported benefits and potential drawbacks of fidget spinners, noting the lack of peer-reviewed evidence supporting their effectiveness for mental health conditions [25]. However, Liang et al. proposed an augmented fidget spinner for biofeedback and respiration training, demonstrating the potential for integrating smart technologies into fidget devices [26].

### 2.2. Blockchain Technology in Healthcare

Numerous researchers have explored the application of blockchain technology in healthcare. Hasselgren et al. provide a comprehensive scoping review of blockchain applications in healthcare and health sciences, highlighting potential benefits in areas such as electronic health records management and clinical trials [18].

Blockchain in healthcare is primarily used for secure data sharing, managing health records, and access control [18,28,40,41]. It addresses the challenges of electronic health records by providing a decentralized and secure method for data exchange. It improves interoperability between disparate healthcare systems, facilitating seamless data exchange and access control [40,41].

Fan et al. proposed MedBlock, a blockchain-based information management system that aims to efficiently and securely share medical data [7]. Their work demonstrates the potential of blockchain in improving the security of data and interoperability in healthcare settings.

Koumpounis and Perry proposed a blockchain-based electronic health record system with patient-centered data access control, highlighting the importance of user empowerment in managing health data [19].

Chowdhury et al. developed a blockchain-based wearable data marketplace to address privacy and trust issues in health data sharing [42], proving blockchain’s potential to improve data security, user control, and privacy in health data sharing from wearable devices.

Dwivedi et al. presented a decentralized privacy-preserving healthcare blockchain for IoT devices, addressing some of the security and privacy challenges associated with health data collection from wearables [5].

### 2.3. User-Centered Design and Technology in Healthcare

The importance of user-centered design in health technologies is well-established. Genaro Motti and Caine provide an overview of wearable applications for healthcare, emphasizing the need for user-centered design approaches to address challenges in user acceptance and adoption [43].

Effective user-centered design involves end-users from the early stages of project development through to final deployment, ensuring their needs and requirements are met [44]. Involving patients in the design and testing phases ensures that health technologies are customized to meet their needs, enhancing functionality and usability [45].

Blockchain technology can be considered challenging from a design perspective [46]. Moniruzzaman et al. underline that the design process of blockchain-based products often lacks systemized practice guidelines [47]. Moreover, designers struggle to understand the particular challenges users face in blockchain-based products [48,49], and developers often cover design-related issues [46].

Despite the growing body of research on blockchain applications in healthcare, several gaps remain. There is limited empirical evidence on the user acceptance of blockchain-based health data-sharing systems. Furthermore, while user-centered design is recognized as important, there is a lack of specific design guidelines for blockchain-based health applications that balance technical requirements with user needs and regulatory compliance.

## 3. Materials and Methods

### 3.1. Iterative Design Process

The development of the smart fidget toy followed an iterative, user-centered design process consisting of multiple phases. Each phase incorporated user feedback to refine and improve the prototype. This approach ensures that the outcome will be closely aligned with user needs and preferences.

Each phase of the design process involved the participation of potential users in various formats (interviews, testing sessions, etc.) and iterative prototyping to refine the device based on direct user feedback.

#### 3.1.1. Initial Concept

The initial concept for the Smart Fidget Toy (Figure 2) was developed based on preliminary research into existing fidget toys and wearable devices for stress management. The goal was to integrate blockchain technology into a device that provides secure data storage while maintaining the tactile satisfaction of traditional fidget toys.

Smart Fidget Toy is proposed as a device to collect data when a stressful moment happens to the user and to correlate it with some health parameters.

The product concept includes the following aspects: shape and ergonomics, sensors and fidgeting elements, possible use cases, and used sensors. The product will be in a round, flat shape to fit comfortably in the user’s palm and enable it to be used as jewelry.

By leveraging blockchain technology, the wearable device aims to provide a holistic approach to mood monitoring, empowering individuals to take proactive steps in managing their mood state and assisting them in their rituals associated with OCD, ADHD, panic attacks, and depression. Smart Fidget Toy, however, is not a medical device and collects biomedical data solely for monitoring and research purposes.

#### 3.1.2. Project Development

To define better the user interactions with the product, we conducted the following studies and exercises (Figure 3):Development of the StoryboardPersona Canvas DevelopmentDevelopment of the Empathy MapComparison of the User Flows of similar productsDevelopment of User FlowsDevelopment of Low-fidelity prototypesApp Map DevelopmentUser Journey Map DevelopmentService Blueprint DevelopmentFocus GroupAnonymous SurveyDevelopment and User Testing of Interactive WireframesDevelopment and User Testing of Mid-fidelity prototype

That approach allowed us to continuously refine the product while maintaining its multidisciplinary development while focusing on a user-centered design approach.

Four of the activities involved potential users. In these studies, the number of participants varied, reflecting both methodological considerations and practical constraints. Our sample sizes are consistent with common practices in user-centered design research, particularly for qualitative data collection and iterative design processes.

For the focus group (*n* = 6), we aimed for a size that would facilitate a discussion while ensuring all participants had an opportunity to contribute. This aligns with recommendations for focus group sizes in design research, which typically range from 4 to 6 participants [50].

The survey (*n* = 28) allowed us to gather a broader range of perspectives, providing quantitative data to complement our qualitative insights. While larger sample sizes are ideal for surveys, this number was sufficient for our exploratory purposes [51].

For the wireframe testing (*n* = 7) and mid-fidelity prototype testing (*n* = 10), these sample sizes are typical for usability studies, where 5 to 10 participants often uncover the majority of usability issues [52,53]. These numbers allowed for in-depth, qualitative feedback on user experience while remaining feasible.

It’s important to note that our study faced time limitations and recruitment challenges, which influenced our final participant numbers. Additionally, to maximize insights from our participant pool, some individuals participated in multiple study phases: two participants were involved in both the focus group and mid-fidelity prototype testing, and two others participated in both wireframe testing and mid-fidelity prototype testing. This overlap allowed us to gather longitudinal insights on user perceptions throughout the design process. While larger sample sizes could potentially offer more detailed results, our approach prioritized deep, qualitative insights at each stage of the design process.

#### 3.1.3. Development of Prototypes

To begin the development of the smart fidget, we created several low-fidelity prototypes focusing either on the technological aspects of the product or on the physical appearance of the device, incorporating basic interaction features.

Those prototypes were developed to validate that the model can be produced, the electronic elements fit inside, and we can test it with users.

The technology-focused prototype was not shown to the audience, but it served as an experimental exercise to define the technologies used for the project. Since the technological aspects significantly influence the design, it was essential to start developing the hardware and software components of the wearable device early in the design process. The draft software to transfer and store the data in the blockchain was developed and connected to a physical low-fi prototype.

The physical artifact was re-developed several times. The 3D model of the prototype was prepared using Grasshopper inside of Rhinoceros [54]. The Grasshopper script describes the design of several separate prototype elements parametrically to allow easier future editing. The following components were developed: the rotary disc, the closing cap to fix the disk, the button cap, and the base. The size of the artifact was influenced by the electronics selected, especially by the height of the rotary encoder and the size of the microcontroller.

The base has supportive elements for the closing cap and rotary disc, a hole for the PPG sensor, another one for the charging cable, and one more for LED. The rotary element is presented as a flat disc with a spherical tip on the top surface and a full-height hole in the center for the rotary encoder. The button cap has a special border to assemble it to the rotary disc and a little tip to fit in the rotary encoder rotating part. The closing cap of the device has a little border to lock the rotary disc inside the device and two holes aligned with the base for the LED and charging cable.

All the elements were fabricated using 3D printing technology, and all elements except for the button cover were printed with white PLA, while for the button, we used different mixes of rubber-like materials.

At this stage, the only electronic element placed inside was the rotary encoder since it provides tangible user feedback.

Those prototypes were developed to verify the selection of electronics and validate their design with potential users.

### 3.2. User Experience Design

Blockchain technology is considered a highly safe data storage method but has a complex user experience [55]. Moreover, most of the microcontrollers used for wearables cannot directly interact with the blockchain, which means that an intermediate layer or bridge solution is required to facilitate communication between the wearable device and the blockchain.

This can introduce additional complexities in designing and implementing blockchain-based wearables for health-related monitoring.

To define the user experience of our product, an analysis of User Flows of the products available on the market was conducted with a focus on user interactions, use of blockchain technology, data collection, and data storage.

We selected several products (Table 1): three blockchain-based products (Patientory is a health app [56,57], and two others are IoT devices—IoTeX Ucam [58] and IoTeX Pebble [59]). We also selected one symptom-tracking app (Bearable [60]), three mental health devices (Apollo [61], TouchPoints [62], and Lief [63]), and one mood companion phygital game (InTempo [64]).

From the analysis of blockchain-based products, we learned that Patientory uses blockchain technology for user authentication, data storage, data access, user rewards, and payments for storage subscriptions [56,57]. IoTeX Ucam uses blockchain for user authentication and data access only [58], while IoTeX Pebble utilizes blockchain for data storage and data distribution [59].

None of the mood/mental health-related products we selected for our analysis utilize blockchain. The physical devices of those products are collecting the data in a different manner. For example, Apollo does not collect the data using the device but rather from the fact of interactions with the product [61], while InTempo tracks user interactions with the device while it is connected to a smartphone and the user plays a game [64]. Lief collects the data from the wearable about users and provides users with biofeedback [63].

This analysis was used to define user flows overall and specifically at what steps of the user flows best to integrate interactions with blockchain technology and how to pass the data from the device to the blockchain.

For our web app, we implemented a similar experience of mood tracking apps for login, records retrieval, and uploading new records interactions. To share the data entries from the device to the blockchain, we decided to use the same web app with a connection to a cryptocurrency wallet for blockchain transactions as commonly implemented in blockchain-based apps.

To limit the efforts of the user, we decided to utilize wired communication between the device and the web app so that the user can charge the device and upload the new data collected by the device simultaneously.

The results influenced the definitions of User Flows, App Map, User Journey Map, and Service Blueprint. The regular user interaction is defined as follows:The user uses the device on demand. The device collects data about user interactions and the user’s heartbeat.The user can connect the device to a web app. If new data is found on the device, the user is advised to upload it to the blockchain network, providing their notes and comments if they want to. Once the data is on the blockchain network, it is permanently deleted from the device.Using the web app, the user can retrieve their previous records from the blockchain. They can only see them if they are logged in to the wallet they use for interactions with the blockchain network.The user can grant or revoke access to their data to doctors and/or researchers. They can get monetary benefits, if applicable, from sharing their data.

### 3.3. Focus Group Study

We conducted a focus group discussion with six individuals from our target demographic (24–32 years old, interested in well-being practices and technology). Participants interacted with the lo-fi prototype and provided feedback on design, usability, and desired features. The feedback collected was used to inform the next iteration of the prototype.

The prototype was presented to a focus group to validate affordances, tactile properties and feedback, ergonomics, the influence of peers on the use of the device, and to adjust the position of the sensor.

The participants were asked to discuss the presented artifact, try using it, and provide their opinions. Then, they were described in more detail about the product and asked more precise questions about tactile feedback, shape, materials, and overall playfulness.

The audio transcript of the study was later coded to identify key issues, key potentials, opinions on material and tactile feedback, and the overall opinion of participants.

To validate the ergonomics and the sensor position, participants were invited to apply a transmittable colored element (eyeshades) on their hands and try to use the device again (Figure 4). Later, we could see the most and least touched spots on the prototype.

Feedback from the focus group highlighted the importance of tactile feedback and ergonomic design, which led to adjustments in the shape and materials used in the next prototype iteration.

### 3.4. Anonymous Survey

An anonymous survey was designed to gather information on the types of data users would find valuable for their well-being practices while using the smart fidget toy. The survey had an exploratory nature and included general demographic questions, questions on user experience with wearables and fidgets, and suggestions for additional features.

The survey was developed as an Airtable [65] form and distributed online, targeting the same demographic as the focus group. We aimed to collect about 30 responses that were analyzed to identify common themes and preferences, which guided the selection of data points to be collected by the smart fidget toy and the web app linked to the device and blockchain network.

The survey had a strict structure; it included both quantitative and qualitative questions, with the quantitative data revealing trends in user preferences and the qualitative data providing insights into desired features. The qualitative data was coded by the themes of the questions in several groups, such as “fidgeting elements” or “important knowledge for the individual”.

### 3.5. Development and User Testing of Interactive Wireframes

Based on the user flows, app map technologies selected, and insights from the focus group and survey, wireframes for the accompanying web application were developed. These wireframes illustrated all key user interactions with the web app:Wallet SetupLinking the Device to the Web AppUpload the Data Collected by the Device to the BlockchainShared Access

Each wireframe was initially developed using Whimsical [66], and later, we added interactivity to it on Figma [67].

To set up the user testing sessions, we created a project on Useberry [68]. The user testing was made of a few main sections:Informed ConsentDemography and Experience QuestionnaireTest TasksPost-testing questionnaire

Seven participants were recruited to test the wireframes through a series of usability tasks. We invited young professionals with experience working in the tech sector or research experience related to blockchain and design to participate. All tests were conducted remotely using Useberry platform [68] in asynchronous modality. Participants were asked to navigate the web application and simulate all four main user flows. They were instructed to provide feedback on the usability, intuitiveness, and overall user experience of the wireframes.

To improve our design decisions in the next stages, we collected the data about interactions user completed and how they navigated through the app to complete the tasks (clicks, misclicks, flows, and time needed) along with their replies to questionnaires.

### 3.6. Development and Testing of the Mid-Fidelity Prototype

At this stage, we developed a mid-fidelity (mid-fi) prototype of the entire product: the device and the web app with the link between them.

#### 3.6.1. Device

The physical device was developed incorporating electronic components to simulate the smart functionalities of the fidget toy. This prototype included Adafruit Qt Py -SAMD21 microcontroller (Adafruit Industries, New York City, NYUSA) with GD25Q16—2 MB SPI Flash in 8-Pin SOIC package (GigaDevice, Beijing, China) soldered on the back side of it, Pulse Sensor (World Famous Electronics LLC, Brooklyn, NY, USA) at the bottom, TSWA-3N-C LFS (C&K Switches, Waltham, MA, USA) rotary encoder with the button, Adafruit Li-Poly 3.7 V 150 mAh battery (Adafruit Industries, New York City, NY, USA) and Adafruit LiIon or LiPoly Charger BFF Add-On for QT Py (Adafruit Industries, New York City, NY, USA).

The 3D model of the device was designed using the Grasshopper plugin inside Rhinoceros [54]. To ensure the precision of the design, we used the drawing of the rotary encoder and PPG sensor provided by manufacturers in the datasheets and the 3D models of all other elements published online by Adafruit [69,70,71,72].

Most of the elements were 3D printed using PLA material, while the tips for fingers were manufactured using a flexible transparent material to provide a comfortable tactile experience for the user, ensuring the transparency of it for the PPG sensor and simplifying the manufacturing process.

The electronic elements were assembled inside of the smart fidget, ensuring that all elements function properly while still providing access to repair its elements.

#### 3.6.2. Software of the Device

The designed code integrates the usage of various components, such as a Pulse Sensor, a Flash Memory module, and a Rotary Encoder. The code is designed to collect data about user interaction and their heartbeat and manage this data through flash memory. The collected data can be recorded, transmitted, or cleared based on the user interaction. The full code can be found on GitHub [73].

The setup() function initializes all hardware components. The device uses Serial Communication for debugging purposes but also for managing the communication with the web app.

The loop() function continuously checks for serial input and manages the system state (COLLECT, SEND, DELETE) based on received commands on a serial port.

The collect() function handles the core data collection logic:Tracks the user interactions with the rotary encoder elements.Reads the heart rate from the pulse sensor, updating the minimum, maximum, and average heart rate values.Monitors the activity timeout to determine the end of a session, storing the data to flash memory when the session ends.

The writeFile() function writes the data as a new session in the JSON file if there was no interaction for the past 5 min. The use of JSON for data formatting ensures that the collected data can be easily transmitted.

#### 3.6.3. Web App

The web application was developed to enable users to upload data collected by the smart fidget toy to a blockchain network. The complete code can be found on GitHub [74].

The web app can be seen in 2 parts:Frontend ApplicationSmart Contract

The frontend application is a user-friendly interface that allows users to interact with their data and the blockchain network. The smart contract is a script deployed on the blockchain network that defines the rules and interactions of the data-uploading process and its storage and retrieval. The overall user-blockchain interaction happens as described (Figure 5):The user uses the fidget as a physical device.The user connects the fidget to their device and opens the web app in the browser.Frontend communicates with a smart contract on the blockchain.

The frontend application was developed using React framework for Javascript [75] and Tailwind CSS [76] for the visual configuration. The app also uses dayjs library [77] and Heroicons [78]. The frontend app followed the styles and design patterns defined by the user testing of the wireframes. We used MetaMask Wallet [79] as a blockchain provider for simplified development and user interaction. The app was deployed using the Netlify platform [80].

The smart contract was implemented using Solidity, a programming language specifically designed for creating smart contracts on the Ethereum blockchain [81], and deployed in Polygon zkEVM Cardona Testnet [82], ensuring the anonymization of the data uploaded. The smart contract manages the data to be written in the blockchain and its format, as well as allowing our app to read the records previously submitted and their details.

The blockchain integration was designed to securely store user data and provide a transparent record of all interactions.

The web app is made of several pages dedicated to different interactions that use smaller components for UI groups and elements to simplify the development and allow the reuse of the components.

To connect the device to the web app, the user has to connect it with the wire to the PC, open the web app in the browser, and ensure that they already have the wallet connected to the selected blockchain network.

The web app uses serial communication to retrieve and delete the data from the device. The UI notifies the user if new data is available on the device. User can decide to upload the data to the blockchain by submitting a form, where they can see the preview of their records and add more notes and comments if needed. Once the data is uploaded to the blockchain, it is permanently deleted from the device.

User can see their previously uploaded records in the calendar. Users can easily see if there were any records for each day, and by clicking on it, they can see more detailed information. The calendar was developed using the tutorial [83]. The app additionally emulates that a researcher requested access to the user’s data.

For the mid-fi prototype, we limited the functionality of the product to the key features: data collection, data uploading, and mocking the data sharing interaction.

#### 3.6.4. Blockchain Integration

To address the challenges of secure data storage and user-controlled health data management, we implemented a blockchain solution using the Polygon zkEVM Cardona Testnet [82]. This choice was motivated by its reduced transaction costs, improved scalability compared to the Ethereum mainnet, and enhanced privacy features, making it particularly suitable for our application.

Using Polygon zkEVM allows our app to benefit from zero-knowledge proofs that significantly increase user privacy [84,85]. zkEVM allows for the execution of smart contracts and transactions without revealing the underlying data, providing a higher level of anonymization than traditional blockchain solutions. This is particularly crucial for health-related data.

The core of our blockchain integration is a Solidity smart contract. This contract manages the storage and retrieval of data collected by Smart Fidget Toy. The contract’s structure is designed to handle both daily summary records and detailed session data efficiently.

The contract utilizes two main data structures:Record: stores daily summaries of fidgeting sessions, including average, minimum, and maximum heart rate data, number of sessions, main fidgeting activity, and total duration.Session: stores detailed information about individual fidgeting sessions, including precise heart rate measurements, duration, timestamps, and user comments and tags.

The data flow from the Toy to the blockchain follows these steps:The device collects session data during user interactions.Data is temporarily stored on the device.A user enters the web app on their device using a browser and connects the fidget toy to the device with a wire.When connected to our web application, the device transfers the collected data.The web application interacts with the smart contract to upload the data to the blockchain.Daily summaries are stored on the blockchain.Individual sessions are stored on the blockchain.Users retrieve data.

Users interact with their data on the blockchain through our web application, which connects to their Ethereum MetaMask wallet. When adding new data, the application calls the appropriate smart contract functions, triggering a transaction that the user must sign. This process ensures that users maintain control over their data uploads.

#### 3.6.5. User Testing of the Mid-Fidelity Prototype

The user testing of the mid-fi prototype was developed to evaluate the usability and effectiveness of the Smart Fidget Toy and its integration with the web app, overall interest in the product, and to understand the value provided by it to the users.

The mid-fi prototype was tested by a group of participants from the target demographic. Users were given the Toy to use in a lab setting, and data was collected during active usage periods. Afterward, participants were asked to upload their data to the blockchain, check their previous records, explore the shared access section, and provide or reject the request via the web app on the laptop provided by the researcher. Users would access the app in the browser and connect the device with the wire to the computer.

Each session was finalized by a semi-structured closing interview, where participants were asked to provide feedback on their experience with the device and web app and their feelings about the data sharing feature and the data control management within the app.

Each session was recorded. The user interaction with the web app was additionally recorded with screen recording.

The audio files were later transcribed and coded into the groups of “errors”, “confusions”, “liked/enjoyed”, and “got curious”. The coded groups were additionally supported with key screens or video recording frames. Metrics such as task completion time, error rates, and user satisfaction were analyzed.

## 4. Results

### 4.1. Focus Group Feedback on Lo-Fi Prototypes

The focus group discussions provided valuable insights into the design, usability, and desired features of the Smart Fidget Toy (Figure 6). Six individuals from our target demographic (24–32 years old, interested in well-being practices and technology) participated.

Key topics covered include perceived usage, ideal size and shape, button placement, texture preferences, data tracking features, and potential stigma around a visible fidget device.

Overall, the prototype received positive feedback, but suggestions were made to smooth sharp edges, add grips, hide seams, and collect additional physiological data. No explicit decisions were made, but action items centered on iterating the design based on ergonomic feedback and testing variations in size, shape, materials, and data tracking capabilities.

Participants appreciated the tactile and interactive nature of the prototype, noting that it provided a satisfying sensory experience. They especially liked the clicking feedback of the button, but they emphasized the lack of feedback of the rotary element.

Users easily understand the purpose of the device. The affordance of the rotating element was clear, though they needed more tips to understand that they also can use the button to fidget it. The item seemed to be a bit too tall. The item also needs rounded edges.

Users noted that the device is quite noticeable and might be subject to stigma. Nevertheless, they admit that people use other items at workplaces without any shame, e.g., softballs. Users became very curious about playing with the fidget. They were trying to get it from the hands of other participants and play more.

Based on the experiment with transmittable material on the palms of participants (Figure 7a), we can see four main contact points between their hands and the device: the tip, “upper part of the surface on the side”, and 40° CW and CCW from it (Figure 7b). We also noticed that few participants naturally placed their fingers at the central point of the bottom of the device. That means that the sensor should be relocated, and additional affordance must be provided.

### 4.2. Survey Results

We collected 28 responses. Most (over 95%) of the survey participants fall in the 25–34-year-old range group, which is our target audience. Most of them work in the creative (over 60%) or tech industries (over 55%). Most of them are quite familiar with the wearables and even use smartwatches daily. 1/3 of participants fidget a few times a day, while an equal number almost never do it.

Most users do not find the heartbeat value crucial while fidgeting, though people who fidget quite often find it more valuable than others (Table 2).

When analyzing the data interests of participants (Table 3), we notice that about 60% of respondents would like to see the data about the duration of the session and the emotional state before and after the session. Many participants also found it relevant to know the intensity of fidgeting, the date and time, specific triggers, and their main activity while fidgeting. At the same time, a tiny number (7.1%) notably expressed interest in the data collection about people around them while fidgeting and environmental factors.

From the qualitative data of the survey, we could collect information on two main topics: what and how people use to fidget and what they would expect from the product.

Most respondents mentioned pens and pencils as their most common fidget; many respondents mentioned using jewelry or simply anything in their proximity. Also, many emphasized that they use their hair or body parts, especially hands/palms/fingers.

### 4.3. User Testing of Wireframes

We invited 8 participants from different countries; 6 of them are 25 to 34 years old with diverse levels of knowledge about blockchain technology.

Most participants managed to complete all tasks and felt confident that they did it correctly. Most users found Shared Access functionality very valuable. 3 out of 4 tasks took relatively little time to complete (up to 40 s) with few clicks required by users to accomplish (Table 4).

#### 4.3.1. Wallet Setup Task

Even though our web app does not include the wallet creation functionality, we wanted to provide our participants with a more immersive experience by asking them to complete the creation of a MetaMask Wallet account and installation of its plugin for the Google Chrome browser.

This process was relatively easy to complete for all users but quite time-consuming (>1 min), with 10–11 clicks to complete.

#### 4.3.2. Link Device Task

Since we have the interaction between the physical and digital worlds, we needed to test if this connection is clear to our users. We asked our participants to imagine that they have a Smart Fidget Toy and to complete a task where they associate their physical device with their MetaMask Wallet. Most users confidently finished this task without any wrong clicks.

#### 4.3.3. Upload Data Task

In this task, users are asked to imagine that it has already collected some data. Users are required to try uploading the data to the blockchain network.

This task seemed much more confusing for the users. From the positions and number of user clicks on the Upload Page, we can see that many participants did not understand why they landed at the data uploading form or where they landed at all after connecting the device to the web app (Figure 8). Based on mouse movement and clicks, we can conclude that the users were trying to reach the form to upload the data while they were already there. The form also seemed complicated to understand, and the steps provided to users were confusing and did not reflect the expected mental model.

Nevertheless, all users completed the task and uploaded their data to the blockchain.

#### 4.3.4. Shared Access Task

This user flow is about a feature provided by the platform that allows others to access user data with their permission. It requires the user to decide how to reach the Shared Access page and reject or decline new requests.

For most users, this task was very straightforward and took the least time to complete, and they took the expected user flow of 2 clicks. Most used the notice message to check shared access, while one used the dashboard navigation. None used an icon in the dashboard.

From this experiment, the overall user experience meets user needs and expectations, but some adjustments are still needed. The Upload Data user flow needs to be redefined and simplified.

From qualitative replies, we can see that potential users would love to know more about the benefits of using them.

### 4.4. Mid-Fi Prototype Testing

We conducted ten user testing sessions among five females and five males. All participants fall in the 25–36 years old age group. They were asked to try to interact with the device (Figure 9) and connect it to a web app (Figure 10 and Figure 11).

All participants were excited to try the device after we demonstrated it to them. When users tried the device, it was noticeable that for some users, the device should be bigger to fit comfortably, but for users with smaller hand sizes, it was not the case. Even though the device itself felt “fragile” to users, they enjoyed the fidgeting interactions with the device, notably the “click” during the rotary fidgeting, which was perceived as “very satisfying”. However, the clicking interaction could be more explicit.

Users also mentioned that they “wish for more feedforward and feedback” while interacting with the device. The affordances on the top part of the device seemed evident to them, while more affordances are needed for the base part.

After the device trial, users were asked to use the companion web app on a laptop. They easily connected the device to the computer using the wire, entering the web app and allowing serial communication between them.

On the interface, they were notified about new data availability for upload, and they followed the procedure. The updated upload form was more intuitive even though some users did not provide further details about their mood state, etc.

The interaction with MetaMask Wallet was easy for users to complete, thanks to the supportive texts in the app. Nevertheless, all participants confirmed they needed more explanations about the elements shown in the MetaMask Wallet popup. Then, users easily explored previous records.

The users found the Shared Access feature very interesting and were motivated to share their data for monetary benefits or easier data sharing with their doctors.

Overall, users expressed interest in the device and the web app. Many of them mentioned that they needed more explanations or supportive materials. For most of them, it was easy to complete all the tasks. From the post-testing interview, all users confirmed unfamiliarity with the blockchain technology. About 30% of participants claimed concern about how their data is collected, stored, and used. Still, most participants in everyday life do not take care of the data they produce or share online. Regarding health-related data, all participants highlighted its importance and their wish to have more control over it. Moreover, one participant mentioned that “considering the amount of data we give to [online platform], when I saw the data it collects since I started to use it a long time ago and underage, I regret there was no one to educate me” (P9) about their data.

## 5. Discussion

### 5.1. Key Findings

The study shows that integrating more technologies into fidget toys can significantly improve their functionality and user engagement. The smart fidget toy developed in this research successfully collected data on user interactions and health-related data, such as heart rate, during usage. The proposed web application helps users to interact with the data and blockchain network. This data can provide actionable insights for users to manage their moods and for healthcare professionals to offer more informed support.

### 5.2. User-Centered Design and Iterative Prototyping

The iterative design process, which included multiple phases of user feedback, was crucial in the development process of the device. Now, Smart Fidget Toy closely aligns with user needs and preferences. Focus group discussions and surveys provided important insights into the desired features and functionalities, such as the importance of tactile feedback, ergonomic design, and the ability to track specific data points like session duration and emotional state. The user testing sessions with interactive wireframes and the mid-fi prototype improved overall control over data and acceptability of the technologies used.

Participants found the device engaging and easy to use but suggested improvements in the feedback mechanisms and additional support materials to enhance the user experience.

### 5.3. Impact of Blockchain Integration

#### 5.3.1. Enhanced Data Security

The decentralized nature of blockchain significantly reduces the risk of data breaches. Unlike traditional centralized systems, where data is stored in a single location vulnerable to attacks, blockchain distributes data across multiple nodes. This makes it more difficult for unauthorized parties to alter or access the data. During our testing, users expressed increased confidence in the security of their health-related data (90%), knowing it was protected by advanced cryptographic techniques inherent to blockchain and anonymization provided by it.

#### 5.3.2. User Control and Privacy

Blockchain technology empowers users by granting them complete control over their data. Users can selectively grant or revoke access to their data, ensuring they maintain privacy and autonomy over their personal information. This feature was particularly well-received by participants, who valued the ability to manage their data-sharing preferences directly. The transparent and user-driven access control model aligns with the growing demand for personalized data privacy solutions.

Our design requires users to manually initiate data transfers by connecting the device to a computer. This approach gives users additional control over when their data is uploaded. It aligns with our goal of enhancing user empowerment and privacy.

#### 5.3.3. User Engagement

The integration of blockchain also suggested improvements to user engagement. For example, users could earn tokens for contributing their data to research studies. 70% of participants found this feature not only easy to use but also beneficial and motivating. This aspect increased user motivation and promoted more sustained interaction with the smart fidget toy, making it a more valuable tool.

#### 5.3.4. Computational Challenges

While blockchain technology offers significant benefits for data security and user control in our Smart Fidget Toy project, it also presents several computational challenges:The Smart Fidget Toy, as a small wearable device, has limited computational power and storage capacity. This constraint necessitated the implementation of a hybrid approach where data is temporarily stored on the device and later uploaded to the blockchain via a web application.We implemented an intermediate web application that bridges the smart fidget toy and the blockchain network. The web application handles all blockchain-related computations when users connect the device to a computer via a wired connection. This includes data serialization, encryption, transaction creation, and communication with the blockchain network.Data collection is optimized to occur only during active user interaction with the device. Limiting data capture to these periods reduces unnecessary processing and conserves the device’s memory and battery life. The collected data is formatted into lightweight JSON files, facilitating efficient storage and quick transfer to the web application upon connection.We chose the Polygon zkEVM Cardano Testnet as our blockchain platform due to its compatibility with Ethereum smart contracts and its efficiency in processing transactions. This network offers lower transaction fees and faster confirmation times than other platforms, reducing the computational load and cost associated with data uploads.In smart contract design, minimizing gas costs and improving overall system performance was crucial. Our iterative development process included multiple rounds of code optimization and gas usage analysis to ensure efficient blockchain interactions.
These challenges required careful consideration in designing and implementing our blockchain-based solution for the smart fidget toy. By acknowledging and addressing these computational aspects, we aimed to utilize the benefits of blockchain technology without compromising the user experience or the practical functionality of the device.

### 5.4. Challenges and Limitations

The development process highlighted several challenges, particularly related to integrating sensor technologies and the complexity of user interactions with blockchain-based systems. The requirement for an intermediate layer to facilitate communication between the wearable device and the blockchain added complexity to the design and significantly influenced design and development decisions. Additionally, the physical design of the fidget toy needed continuous adjustments to balance the inclusion of electronic components with user comfort and tactile satisfaction.

This study included only limited interactions with the device and web application. User testing sessions were conducted with a task list provided to the participants. Future research should explore the long-term effects of using blockchain technology for health data. Additionally, further refinements in the device’s design and functionality, including more advanced sensor integration and enhanced user interaction features, will be necessary to improve its effectiveness and user satisfaction.

## 6. Conclusions

The Smart Fidget Toy developed in this study represents an advancement in wearable technology for the health sector using blockchain. Combining user-centered design techniques and the security features of blockchain technology, the device offers a novel approach to mood, stress, focus management, data collection, storage, and sharing. The iterative design process and continuous user feedback ensured that the product met the needs and expectations of its target audience. Moreover, it helped to improve user acceptance of the new technologies, making it not only a valuable tool for users, healthcare providers, and researchers but also providing insights on the design decisions to be taken.

To sum up, similar projects can follow the development process as described:Concept Development
Initial Research: Investigate existing products and identify gaps where technology can improve functionality.Define Objectives: Outline the specific goals of the product, focusing on user needs and potential user benefits.Comparative Analysis: Analyze similar products to refine user flows, data interaction, and overall experience.Technology Integration: Research security and implementation challenges of the suggested technologies.Iterative Design Process
Low-Fidelity Prototyping: Start with basic prototypes involving stakeholders to shape the initial concept and get early feedback on the physical and technological aspects.User Feedback: Conduct focus groups, user testing sessions, and surveys to gather early insights on usability, desired features, and ergonomic considerations.Refinement: Continuously improve the design based on user feedback.
Blockchain Integration
Data Security: Make sure that the user data is securely stored and transmitted in the most secure manner.User Control and Transparency: Enable user-controlled data access and transparency in data transactions and access history to build trust and ensure privacy.

Key Considerations:User-Centered Focus: Maintain a strong focus on the user’s needs and preferences throughout the project to ensure the product is intuitive and meets the intended needs effectively.Technology Suitability: Evaluate and select technologies for their innovation and practical benefits in terms of security, privacy, and user engagement.Iterative Development: Leverage iterative design and testing processes to continually improve the product based on actual user interactions and feedback.Stakeholder Engagement: Engage with all stakeholders, including potential users, healthcare professionals, and technology experts, throughout the development process to align the product with broader system requirements and user expectations.

## Figures and Tables

**Figure 1 sensors-24-06582-f001:**
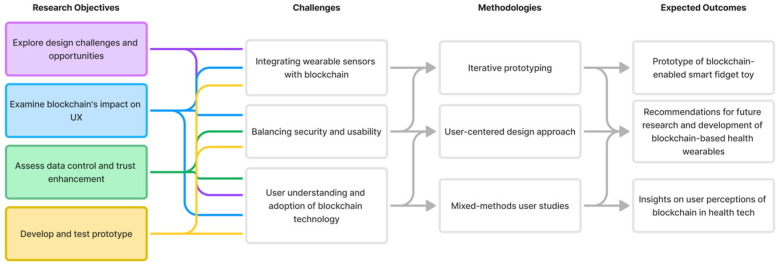
Overview of Research Objectives, Challenges, Methodology, and Expected Outcomes.

**Figure 2 sensors-24-06582-f002:**
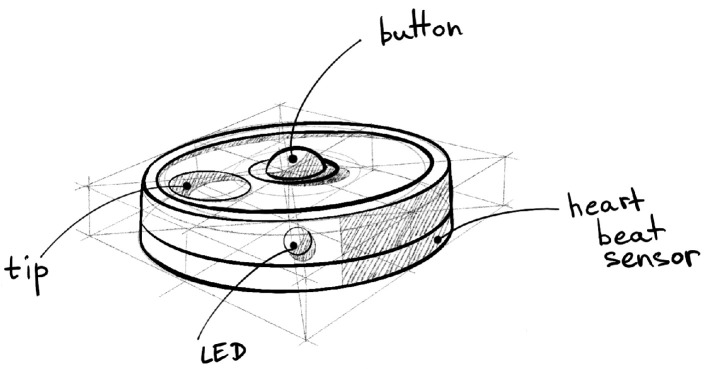
Sketch of Initial Concept of the Device.

**Figure 3 sensors-24-06582-f003:**
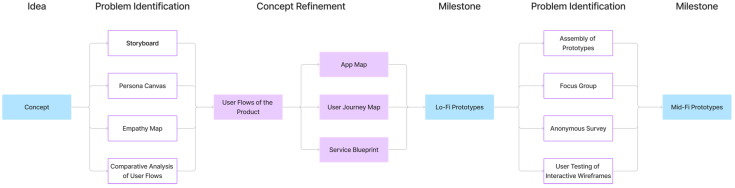
The Development Process of the Smart Fidget Toy.

**Figure 4 sensors-24-06582-f004:**
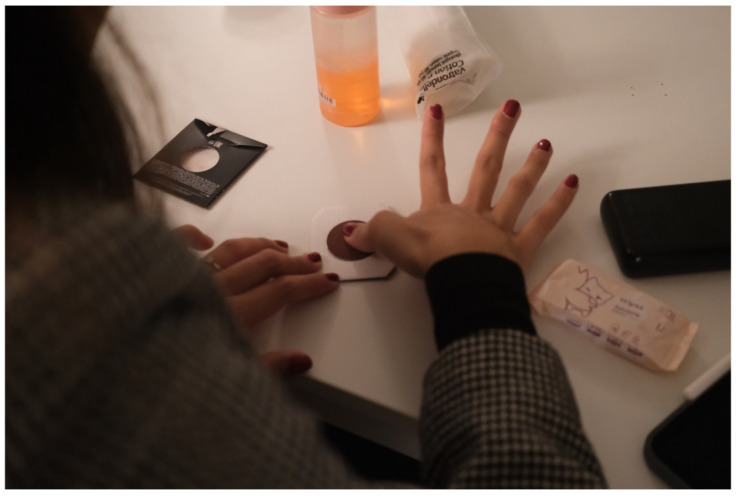
Participant Applying Transmittable Material on their Hands for an Experiment.

**Figure 5 sensors-24-06582-f005:**
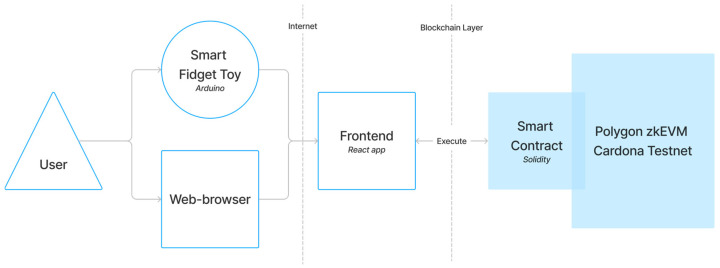
Architecture of the Web app.

**Figure 6 sensors-24-06582-f006:**
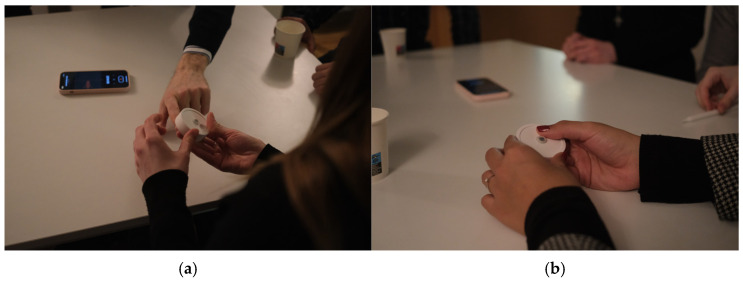
Low-Fidelity Prototype during Focus Group: (**a**) participants discussing the low-fidelity prototype; (**b**) participants trying to use the low-fidelity prototype.

**Figure 7 sensors-24-06582-f007:**
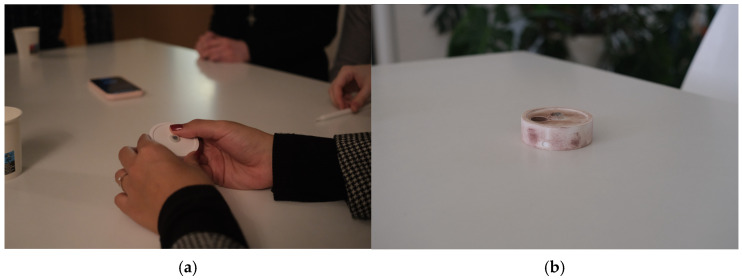
Experiment during Focus Group with Transmittable Material on Hands of Participants: (**a**) participant trying the prototype with transmittable material on their hand; (**b**) low-fidelity prototype with marks left by participants’ trials after the completion of the study.

**Figure 8 sensors-24-06582-f008:**
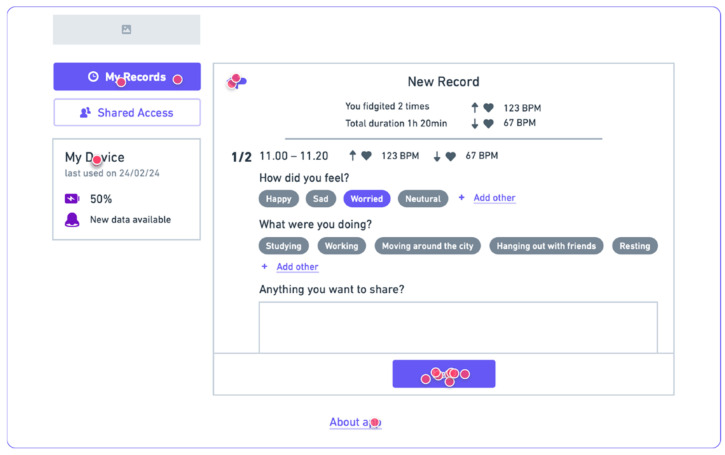
User Clicks on Upload Page on the Web App.

**Figure 9 sensors-24-06582-f009:**
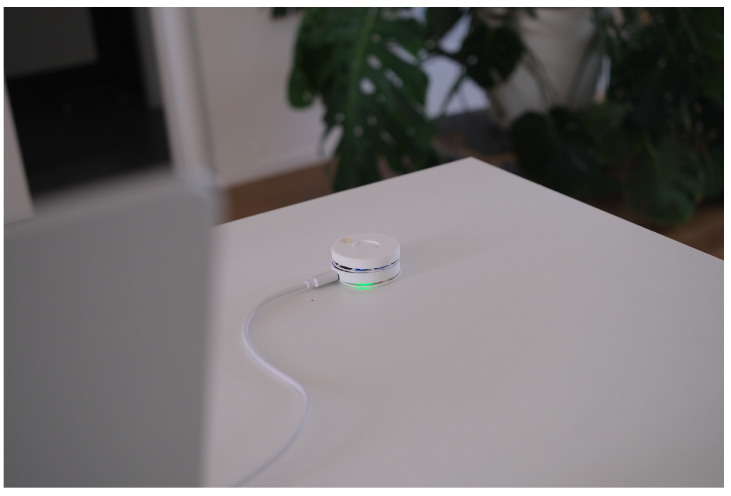
Mid-fidelity Prototype of Smart Fidget Toy.

**Figure 10 sensors-24-06582-f010:**
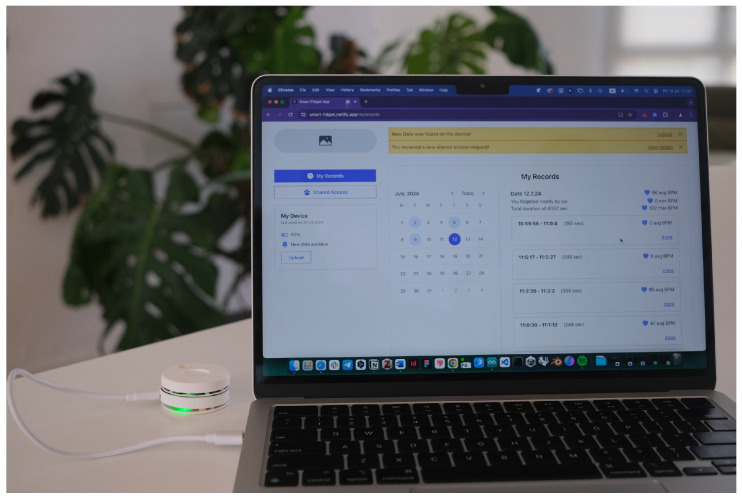
Mid-fidelity prototype of Smart Fidget Toy connected to a Web App.

**Figure 11 sensors-24-06582-f011:**
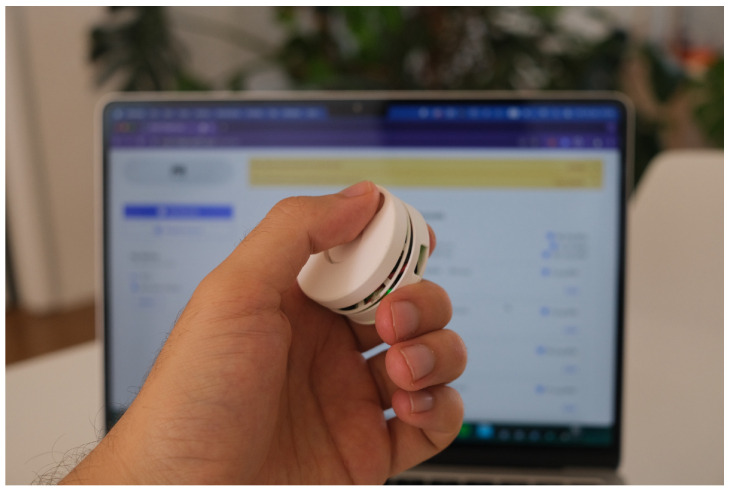
User Testing Session of Mid-fidelity Prototype.

**Table 1 sensors-24-06582-t001:** User Flow Analysis of Selected Products.

Product	Product Category	Use of Blockchain	Data Entry
Patientory [38,39]	Health App	For User Authentication, Data Storage, Data Access, User Reward, Subscription Payment	User Entry, Healthcare Providers, Third-Party Products
IoTeX Ucam [40]	IoT	For User Authentication, Data Access	From the device
IoTeX Pebble [41]	IoT	For Data Storage, Data Distribution	From the device
Bearable [42]	Mood Tracking App	No	User Entry, Third-Party Products
Apollo [43]	Mental Health Device	No	From the use of the product
Touchpoints [44]	Mental Health Device	No	No
Lief [45]	Mental Health Wearable	No	From the wearable
InTempo [46]	Mood Companion (Device + App)	No	From the user interactions with the device and/or app

**Table 2 sensors-24-06582-t002:** Survey Responses to “How important is it for you that the smart fidget device tracks your heart rate?”.

Option	Number of Participants Selected	% of Participants Selected
Very important	1	3.6
Somewhat important	13	46.4
Not very important	8	28.6
Not important at all	6	21.4

**Table 4 sensors-24-06582-t004:** Overall Quantitative Results of User Testing of Interactive Wireframes.

Test Task	Average Time	Median Number of Clicks	Min Number of Clicks	Max Number of Clicks
Wallet Setup	64.3 s	10	10	11
Link Device	12.4 s	2	2	4
Upload Data	36.9 s	8	6	16
Shared Access	14.5	3	2	7

**Table 3 sensors-24-06582-t003:** Survey Responses to “What other metrics or data would you find valuable for the smart fidget to track and analyze?”.

Metrics	Number of Participants Selected	% of Participants Selected
Duration of fidgeting sessions	16	57.1
Intensity of fidgeting sessions	15	53.8
The date and time of the fidgeting session	13	46.4
Environmental factors	9	32.1
Emotional state before and after fidgeting	16	57.1
Specific triggers for fidgeting sessions	13	46.4
People around you while fidgeting	2	7.1
Your main activity while fidgeting	13	46.4
The location while fidgeting	10	35.7

## Data Availability

The data presented in this study are available at the request of the corresponding author (for privacy, legal, and ethical reasons).

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
