# Peer review of "Design and Development of a Smart Fidget Toy Using Blockchain Technology to Improve Health Data Control"

_sensors, 2024, doi:10.3390/s24206582_

Round 1

Reviewer 1 Report

Comments and Suggestions for Authors

The design and development of a smart fidget toy that uses blockchain technology to allow for user-controlled health data management is covered in this study. The incorporation of blockchain technology into health devices is becoming more and more crucial for safe data handling as worries about data privacy increase. With the help of this Smart Fidget Toy, users can keep tabs on their mood swings and fidgeting patterns. They can also securely upload this data to a web application that runs on blockchain technology, providing them complete control over their personal health data. The overall presentaion of the paper is good. But there are few suggetion which will improve the paper:

1. The authors need to rewrite the abstract primarily to highlight the objectives, authors' contributions, and critical research findings.

2. The paper's organization, particularly in Sections 1 and 3, needs improvement.

3. Section 1 also contains subsection 1.1 Background and Motivation; authors must rearrange the sections and subsections.

4. In the introduction section, authors need to include a basic figure that will outline the research objectives and current challenges.

5. The literature section is missing; how can authors determine or validate the novelty of their contribution to the problem statement? They need to conduct a thorough review of existing literature to identify research gaps and identify their own unique contributions.

6. How do authors address the computational challenges or complexity of blockchain technology?

7. The authors mentioned that they chose a different number of participants (such as 7/8/10) for their experiment. Is there any specific meaning to taking a different number of participants?

8. The authors mentioned "Scheme 1: Architecture of the Web App" on page number 10, but figure number should be mentioned instead.

9. Blockchain integration with the proposed scheme is insufficient. The authors did not mention how they used or implemented blockchain in their work.

10. It would be more comprehensible if the authors presented their results in tabular form or in a figure that included all these statistics. The authors may include comparative analysis with existing research.

Comments on the Quality of English Language

NA

Author Response

  1. Comment: The authors need to rewrite the abstract primarily to highlight the objectives, authors' contributions, and critical research findings.
    Response: The abstract has been revised to clearly state the research objectives, our key contributions, and the critical findings from our study. We now highlight our iterative design process, the development of a blockchain-based fidget toy prototype, and the insights gained from user studies regarding blockchain integration in health wearables.
  2. Comment: The paper's organization, particularly in Sections 1 and 3, needs improvement.
    Response: We have restructured Sections 1 and 3 to improve clarity and logical flow. The introduction now provides a clearer roadmap of the paper, and the methodology section has been reorganized to better reflect our iterative design process.
  3. Comment: Section 1 also contains subsection 1.1 Background and Motivation; authors must rearrange the sections and subsections.
    Response: We have adjusted the structure of Section 1, separating the background and motivation into distinct subsections to improve readability and logical progression of ideas.
  4. Comment: In the introduction section, authors need to include a basic figure that will outline the research objectives and current challenges.
    Response: We have added figure (Figure 1) to the introduction section, which provides a visual overview of our research objectives, challenges, and methodology. This figure is accompanied by explanatory text to improve understanding.
  5. Comment: The literature section is missing; how can authors determine or validate the novelty of their contribution to the problem statement? They need to conduct a thorough review of existing literature to identify research gaps and identify their own unique contributions.
    Response: We have added a literature review section (Section 2) that covers wearable devices in healthcare, blockchain technology in healthcare, and user-centered design and technologies in health.
  6. Comment: How do authors address the computational challenges or complexity of blockchain technology?
    Response: We have added subsection 5.3.4 in the discussion section to address the computational challenges of integrating blockchain technology with our wearable device. This subsection outlines our approach to handling limited device resources, data transfer, and blockchain interaction.
  7. Comment: The authors mentioned that they chose a different number of participants (such as 7/8/10) for their experiment. Is there any specific meaning to taking a different number of participants?
    Response: We have added explanations in section 3.1.2 regarding our participant selection. The varying numbers reflect both methodological considerations and practical constraints in recruitment. We also clarify how these sample sizes align with common practices in user-centered design research.
  8. Comment: The authors mentioned "Scheme 1: Architecture of the Web App" on page number 10, but figure number should be mentioned instead.
    Response: We have corrected this inconsistency. The scheme is now referred to as "Figure 4: Architecture of the Web App" throughout the paper.
  9. Comment: Blockchain integration with the proposed scheme is insufficient. The authors did not mention how they used or implemented blockchain in their work.
    Response: We have added a detailed section in the methodology (Section 3.6.4) that explains our blockchain implementation. This section covers the choice of blockchain platform, smart contract design, and data flow between the device, web application, and blockchain.
  10. Comment: It would be more comprehensible if the authors presented their results in tabular form or in a figure that included all these statistics. The authors may include comparative analysis with existing research.
    Response: We have added tables (Table 2, 3 and 4) to present quantitative data from our user studies, particularly in sections 4.2 and 4.3.

Reviewer 2 Report

Comments and Suggestions for Authors

The paper presents a user-centered development of a blockchain-powered device to process user medical/health data.

The paper is easy to read and shows that the authors are highly skilled software developers. They use user-centered methodology to ensure a high user experience and the solution's usability. Surveys and testing were conducted.

Despite its strength, the section architecture needs revision. I cannot figure out how to access/reach the front end using the fidget toy. Scheme 1 and Fig. 6 need explanation. Which parameter of the heatmap is assessed? What do the colors mean?

Author Response

  1. Comment: I cannot figure out how to access/reach the front end using the fidget toy.
    Response: We have clarified the interaction between the physical device and the web application in section 3.6.3. We explain that users connect the device to their computer via a wired connection and then access the web application through a browser.
  2. Comment: Scheme 1 and Fig. 6 need explanation.
    Response: We have added more detailed explanations for Figure 4 (previously Scheme 1) in section 3.6.3. The Figure 8 (previously Figure 6) was modified and accomplished with more detailed explanation.
  3. Comment: Which parameter of the heatmap is assessed? What do the colors mean?
    Response: We have modified Figure 8 (previously Figure 6) by replacing heatmap representation with clicks on the interface and adjusted the accompanying text next to it.

Round 2

Reviewer 1 Report

Comments and Suggestions for Authors

The authors address my concerns. No more comments.